# Single-Cell RNA-Sequencing Analysis Provides Insights into IgA Nephropathy

**DOI:** 10.3390/biom15020191

**Published:** 2025-01-29

**Authors:** Ming Xia, Yifu Li, Yu Liu, Zheng Dong, Hong Liu

**Affiliations:** 1Department of Nephrology, Hunan Key Laboratory of Kidney Disease and Blood Purification, The Second Xiangya Hospital, Central South University, Changsha 410011, China; xiaming04@csu.edu.cn (M.X.); liyifu@csu.edu.cn (Y.L.); rory0423@csu.edu.cn (Y.L.); zdong@augusta.edu (Z.D.); 2Center for Medical Research, The Second Xiangya Hospital, Central South University, Changsha 410011, China; 3Department of Cellular Biology and Anatomy, Medical College of Georgia at Augusta University, Augusta, GA 30912, USA; 4Charlie Norwood VA Medical Center, Augusta, GA 30901, USA

**Keywords:** IgA nephropathy, single-cell, RNA-sequencing

## Abstract

IgA nephropathy (IgAN) is currently the most common primary glomerular disease worldwide, and early diagnosis and intervention contribute significantly to improving outcomes and reducing the incidence of renal failure. The pathogenesis of IgAN remains incompletely understood. In recent years, the rapid development of single-cell RNA-sequencing (scRNA-seq) technology has provided the high-resolution and rich data necessary to elucidate disease characteristics and enabled the analysis of complex interactions between individual cells and cell types. The application of scRNA-seq in IgAN successfully revealed the landscape of immunological features, including peripheral blood B-cell and Th-cell activation, cytotoxic T-cell depletion, and renal infiltrating cell subtypes, as well as the crucial role of mesangial cells and endothelial cells in the early stage of kidney injury, and also revealed the extensive intercellular interactions between renal cells. Here, we discuss the research progress of scRNA-seq technology in IgAN. These scRNA-seq profiles help us to understand the complex molecular mechanisms of IgAN and develop biomarkers and specific therapeutic strategies.

## 1. Introduction

IgA nephropathy (IgAN), also called Berger’s disease, presents as a chronic renal disorder characterized by the deposition of IgA in the glomeruli. It is currently the prevailing form of glomerulonephritis globally. Its prevalence is modest in the United States (~20%), higher in some European countries (20–30%), and highest (~40%) in Asia, especially in China (54%) [1]. IgAN mainly occurs in young and middle-aged people, and its clinical phenotypes exhibit marked heterogeneity, ranging from asymptomatic microscopic hematuria to rapidly progressive glomerulonephritis. Common clinical manifestations include hematuria, proteinuria, hypertension, and declining renal function [2]. The prognosis of IgAN varies widely: some individuals maintain stable renal function for a few decades, while around 30% of patients experience a progression to end-stage renal disease (ESRD) within a relatively short timeframe (4–15 years) [3]. The exact pathogenesis of IgAN is incompletely understood and is currently thought to be the result of a complex interaction among genetic predisposition, the microbiome, immune dysregulation, and environmental factors [4].

The status of IgAN on the global health stage is increasingly rising. Although gene microarrays and high-throughput sequencing have provided important gene expression data on IgAN [5,6], they are unable to effectively capture the cellular heterogeneity and spatial structure of the disease. Specifically, these technologies are unable to accurately identify gene expression differences across various cell types, limiting their ability to provide deeper insights into the specific roles of different cell types, cellular states, and their interactions, which limits the understanding of the dynamic changes during disease progression. Single-cell RNA sequencing (scRNA-seq) is an innovative technique that allows for the sequencing of the transcriptome at the individual cell level [7]. Its strengths lie in its ability to analyze gene expression and cell–cell communication, assess functional enrichment, explore metabolic pathways, and discover new cell subtypes within heterogeneous populations. Because it can tackle the challenges of limited sample sizes and cellular diversity, scRNA-seq has been used to study several kidney diseases, including IgAN, and has become a potent tool for deepening our understanding of the disease onset and progression. ScRNA-seq has identified marker genes for different components of the renal vasculature and distinguished cell subtypes (more than 34) of various functional units of the kidney [8]. The following sections provide an overview of the mechanisms of IgAN and the new understanding that the scRNA-seq technique has brought about.

## 2. General Mechanisms/Pathophysiology of IgAN

IgAN has been proposed to be affected by genetic, environmental, and microbial factors, among others, and to develop through a ‘four-hit’ process. (1) On the basis of genetic susceptibility, individuals produce galactose-deficient IgA1 (Gd-IgA1) after mucosal immune responses are triggered by bacterial or viral infections. Genome-wide association studies (GWASs) have identified over 30 risk loci for IgAN [9], but, so far, they only explain about 11% of heritability [10]. For some susceptibility genes, it is still unclear which cells are predominantly affected, as well as their main function under IgAN circumstances. The gut and tonsils are considered important sites for antigen recognition and immune cell activation. Pathogenic microorganisms provoke T-cell-independent (stimulated by epithelial, dendritic, and stromal cells through the generation of cytokines such as IL-6 and IL-10, TGF-β, BAFF or BLyS, and APRIL) and T-cell-dependent mechanisms (stimulated by TLR-mediated BAFF secretion and MHCII expression on B cells), thus inducing B-cell differentiation and IgA production [11]. It is believed that the expression or activity of glycosyltransferase and the species of colonized mucosal microorganisms are associated with the aberrant glycosylation of IgA1 [12]. Notably, the glycosylation deficiency is specific to circulating IgA1 but not to other glycoproteins in IgAN patients, suggesting that Gd-IgA1 production may be limited to a specific subset of B lymphocytes. The subsequent hits are (2) the production of anti-Gd-IgA1 antibodies, including IgM, IgG, or IgA [13]; (3) the formation of immune complexes containing Gd-IgA1, anti-glycan antibodies, and various other biological proteins, such as sCD89 shedding from mononuclear/neutrophils [14]; and (4) the deposition of immune complexes in the renal mesangium, leading to complement system activation, mesangial matrix expansion, and inflammation, and ultimately resulting in glomerular sclerosis and interstitial fibrosis. Various IgA receptors on mesangial cells (MCs) have been identified (transferrin receptor; integrins a1β1, a2β2, and β-1; and 4-galactosyltransferase 1) [15], and evidence suggests that the alternative pathway and lectin pathway are involved in IgAN [16]. However, in some cases, IgA deposition does not cause changes in glomerular morphology [17], while in other cases, severe inflammation, including crescent formation, occurs. The inter-individual heterogeneity of IgAN remains to be elucidated. 

As discussed above, several key issues in the pathogenesis of IgAN are unclear, including the specific effects of susceptibility genes on certain cells, the characteristics of B-cell subsets, the landscape of immune cells, the initiator of kidney injury, and the interplay of biological events after immune complex deposition. Further exploration of these issues will help us to understand the occurrence and progression of IgAN.

## 3. Development of scRNA-Seq Technique and Its Application in Nephrology

ScRNA-seq technology emerged in the early 2010s [18], but the initial approaches faced technical limitations, including low sensitivity and high technical complexity. With advancements in microfluidics, molecular barcoding, and bioinformatics, several platforms and sequencing technologies have been developed for scRNA-seq [19]. At present, there are two main methods for single-cell separation. The microplate approach evolved from the initial STAT-seq, SMART-seq, and Fluidigm C1 to STRT-seq-2i and BD Rhapsody, in which individual cells are deposited into separate micropores and labeled. The droplet-based method, which separates and recognizes cells by constructing fine microdroplets, evolved from DropSeq and InDrops to commercial 10× Chromium, offering scalable and cost-effective solutions for single-cell profiling [20,21]. Open-source tools such as Seurat, STAR, and HISAT2 offer greater flexibility and customization over preprocessing, quality control, and alignment tasks for scRNA-seq analysis. Seurat, for instance, is widely used for clustering and differential expression analysis, while STAR and HISAT2 excel in aligning RNA-seq reads to reference genomes, particularly in detecting splice variants [22]. Those bioinformatics programs play a crucial role in improving data quality by excluding low-quality cells and genes, and enhancing the accuracy of downstream analyses such as cell-type identification and gene expression profiling.

The procedure of scRNA-seq can be summarized into four steps: cell sorting and capture, reverse transcription, cDNA amplification, and sequencing library preparation [23]. The workflow and prevalent methods, encompassing raw data processing, quality control, etc., have been outlined previously [24]. The application of scRNA-seq technology in nephrology provides important tools and insights for understanding renal pathogenesis and developing therapeutic strategies. In particular, IgAN specimens for scRNA-seq are derived from the kidneys or peripheral blood, as described in the reviewed literature. For renal samples, tissues are typically dissected into millimeter-sized fragments and digested utilizing collagenase and trypsin. Blood samples are usually suspensions of peripheral blood mononuclear cells (PBMCs) obtained via Ficoll-Paque density gradient centrifugation and can be isolated by cell sorting in advance according to the experimental demand. ScRNA-seq can identify distinct cell populations according to their gene transcription characteristics, as well as facilitate the elucidation of cell dynamics in response to various stimuli or different disease states, providing valuable insights into the pathological mechanisms and precision medicine treatment approaches.

## 4. Overview of Research on IgAN Using scRNA-Seq

The first study of IgAN scRNA-seq was published in 2020 [25], specifically highlighting the aberrant expression of genes associated with IgA deposition in MCs as a trigger of IgAN pathogenesis, leading to tubulointerstitial inflammation and fibrosis. To date, seven more studies have been published, as shown in Table 1, which not only revealed the critical role of kidney innate cells such as MCs and endothelial cells (ECs) in the development of IgAN but also identified the genetic changes in and characteristics of different immune cell subpopulations, providing insights into new IgAN biomarkers, diagnoses, and treatments. The following subsections describe the key information conveyed by the scRNA-seq of potentially pathogenic cells in IgAN.

### 4.1. Recognition of Peripheral Blood Cell Types and Signals via scRNA-Seq

#### 4.1.1. Peripheral Blood B Cells

B cells are the source of Gd-IgA1 and its antibodies in IgAN. ScRNA-seq data from Zeng H et al. [27] hinted at viral influences in IgAN pathogenesis. The authors divided B-cell clusters into four subgroups, of which three expressed naive B-cell markers and one expressed high levels of JCHAIN and AIM2, which are mainly expressed in memory B cells. HLA-C is a major histocompatibility complex (MHC) class I molecule that is consistently elevated in all B-cell populations. Furthermore, CD83 and CD69 were upregulated, indicating increased B-cell activation and a stronger antigenic response in IgAN. One of the subsets with suppressed NF-κB signaling showed a predominant presence in IgAN and was positively associated with renal severity markers (urine protein-to-creatine ratio, urine erythrocyte, galactose-deficient IgA1, IgA, and C4), suggesting its involvement in IgAN progression. Chen Q et al. [31] defined six B-cell subtypes, and their results suggested that plasma cells at the end of differentiation were increasingly dominant in the disease. In particular, B cells showed greater activation of gene expression related to IgA and IgG production in nephrotic syndrome (NS)-IgAN than healthy controls [31].

#### 4.1.2. Peripheral Blood NK Cells and T Cells

T cells participate in immune tolerance, secrete proinflammatory cytokines, assist B lymphocyte differentiation, and produce autoantibodies in the pathogenesis of IgAN [33]. Natural killer (NK) cells, as the third largest lymphocyte group after T cells and B cells, are considered the bridge between innate and adaptive immunity. IgAN scRNA-seq results showed that NK cells exhibited functional impairment and exhaustion, characterized by reduced activation and cytotoxic functions. This diminished activity was reflected in the significant downregulation of effector molecules, such as PRF1 and CCL4. Moreover, NK cells demonstrated the upregulation of FOSB and JUNB, genes encoding components of the AP-1 complex, and HLA-C, a key MHC class I receptor, which influences NK cell cytotoxicity. Subsets with high levels of CX3CR1, or IL2RB, and FCER1G were significantly reduced in IgAN. These specific expression patterns are probably due to chronic infection and/or inflammation and affect immune cell proliferation and modulation [27].

Notably, Tfh cells (CXCR5^+^PD-1^+^), Th2 cells (CCR4^+^), and Th17 cells (CCR6^+^) were significantly increased in IgAN patients, and the proportion of Tfh cells was positively correlated with kidney injury markers (such as 24 h urinary protein and the urinary protein–creatinine ratio). In addition, Tfh cells were found to exhibit an enhanced ability to support B-cell responses by regulating CD40-CD40L signaling or genes linked to B-cell differentiation [30]. Effector Tregs expressing CCR4 may play a role in kidney injury resolution in NS-IgAN [31], whereas the proportions of Treg cells (CD25^+^CD127^−^) and Th1 cells (CXCR3^+^) showed no significant differences compared to healthy controls [30]. 

#### 4.1.3. Peripheral Blood Monocytes

Monocytes are identified as classical (CD14^high^, FCER3A^neg^), intermediate (CD14^high^, FCER3A^low^), and non-classical monocytes (CD14^low^, FCER3A^high^). The classical type can be further subdivided (into at least four subsets) in IgAN. Several key markers, including HLA-B/C, TMEM176A/B, and NAMPT, were significantly increased, and their expression levels were positively correlated with clinical parameters of clinical severity. Strictly speaking, HLA-B and HLA-C are associated with antigen presentation, NAMPT plays a role in inflammation and metabolism, and TMEM176A/B suggests the potential modulation of inflammasome activity. Thus, it can be inferred that IgAN monocytes possess an enhanced antigen presentation capacity. MHC class II molecules and oxidative phosphorylation genes were also increased in intermediate monocytes of NS-IgAN patients [31]. In contrast, decreased levels of chemokines such as CCL3, CCL4, CCL3L3, and CCL4L2 were observed in IgAN, suggesting impaired monocyte-mediated chemotaxis. Classical monocytes could self-regulate and showed upregulated interferon signaling independent of NK cell function in IgAN [27]. In addition, the transcriptional regulator KLF4, predominantly upregulated in both classical and non-classical monocytes, was reported to play a pivotal role in controlling the monocyte transcriptional network and differentiation [28].

Moreover, monocytes/macrophages in IgAN patients showed high expression of M1 macrophage markers and C3. Pseudotime analysis revealed that the differentiation trajectory of blood monocytes to kidney tissue macrophages was accompanied by the upregulation of immune and fibrosis pathways [32]. 

### 4.2. Identification and Characterization of Cell Types in Renal Tissue Using scRNA-Seq

#### 4.2.1. Mesangial Cells

Mesangial cells (MCs) provide central structural support to the glomeruli. These cells exhibit dysregulated extracellular matrix (ECM) dynamics and heightened inflammatory signaling, integrin binding, and iron channel binding in IgAN. Notably, JCHAIN, WFDC2, and FN1 are significantly elevated, suggesting a role in IgA polymerization and ECM accumulation. The increased expression of KNG1, PLGRKT, AOC3, and immune regulatory cytokines such as CXCL12, IL34, CSF1, CCL2, CCL3, CCL4, and CXCL1 in MCs contribute to inflammatory responses and immune cell recruitment [26,32]. MCs also express growth factors and signaling pathways involved in mesangial proliferation and fibrosis with the overexpression of specific genes, such as MALAT1, GADD45B, and FOS [26]. In comparison to microproteinuria patients, IgAN patients with overt proteinuria had MCs enriched in genes participating in complement activation and the alternative pathway, with high levels of the ECM regulation protein SPARC and ROCK2 [26]. The significance of genes such as CLIC1, ribosomal subunit protein RPS26, SOX4, and EDIL3 is still unclear, necessitating further investigation [28]. 

In gddY mice used as animal models, scavenger receptors (Cd36, Stab2) involved in IgA/immune complex internalization were upregulated [29]. MC proliferation, along with significant immune cell infiltration, was observed during IgAN progression. Distinct MC subtypes (proliferative and inflammatory MCs) were identified, and inflammatory MCs possibly originated from proliferative MCs. Targeting the Cxcl12/Cxcr4/C3 signaling pathway has shown potential in attenuating inflammatory injury, fibrosis, and renal function decline in IgAN [32].

#### 4.2.2. Endothelial Cells

Endothelial cells (ECs) are essential for glomerular filtration and can be annotated to three major subtypes: glomerular, afferent arteriolar, and efferent arteriolar ECs. In IgAN, differentially expressed genes in ECs include overexpressed genes involved in cellular adhesion (SELP and PECAM1), angiogenesis (XIPS), EMT (SOX4), leukocyte recruitment (ACKR1), and homeostasis (RNASE1). In patients with overt proteinuria, ECs showed increased expression of genes involved in ECM binding and cell–substrate adherence junctions [26]. The scRNA of gddY mice also uncovered a role for ECs in immune cell recruitment and identified several intraglomerular paracrine pathways that promote inflammation. The results showed the upregulation of genes associated with endothelial dysfunction (e.g., Edn1, f8, Sele, Vcam1, Icam1, Spp1, Cd36, and Nostrin) and proinflammatory responses (Cx3cl1, Cxcl1, Cxcl11, and Serpina 3n, h, e, f, and g). Notably, MHC class IB genes, known to mediate immune responses in ECs, were significantly overexpressed, while many MHC class II genes were downregulated [29].

#### 4.2.3. Podocytes

Podocytes are glomerular epithelial cells and contribute to the maintenance of the glomerular basement membrane by secreting collagen. They have been postulated to serve as a crucial hub for communication in the kidney under basal conditions and in response to cellular stress [34]. In scRNA-seq of IgAN samples, several key factors were observed to be elevated. These include PRSS23, a serine protease implicated in renal fibrosis; NGF, which is known for its protective role in the kidney; and HES1, involved in Notch signaling and EMT [26]. The characteristics of the EMT of podocytes were significantly prominent in NS-IgAN patients [28] and animal models. Proinflammatory and profibrotic genes such as B2m, H2-k1, Col1a2, and Acta2 were upregulated in gddY mice [29]. In addition, CD74, B2 M, and FXYD5 were identified as potential therapeutic targets in podocytes [28].

#### 4.2.4. Uriniferous Tubule Cells

On the basis of available techniques, urinary tubules are considered to consist of renal tubules (proximal renal tubules, Henle’s loop, distal renal tubules) and collecting tubules. The proximal renal tubule mainly plays a role in reabsorption; scRNA revealed that the traditional subtypes of proximal tubule cells (S1, S2, and S3) showed no significant differences in the expression of key markers. The number of proximal tubule cells in IgAN samples was reduced compared to that in healthy donors, indicating their susceptibility to nephritis-related injuries [28]. Moreover, genes involved in TNF, IL-17, and NOD-like receptor signaling and the regulation of leukocyte transendothelial migration are upregulated in proximal tubule cells. Leukocyte transendothelial migration, chemokine signaling, and the type I interferon pathway were increased in the proximal tubule cells of IgAN subjects with overt proteinuria compared to those with microproteinuria [26]. In the gddY IgAN model, genes associated with fibrosis, inflammation, and the response to injury, including the upregulation of C3, showed the most significant changes [29]. In distal tubule cells, genes related to the MAPK cascade and p38MAPK cascade were enriched in IgAN.

The loop of Henle, principal cells, and intercalated cells in IgAN were enriched for genes involved in TNF signaling, IL-17 signaling, Th17 cell differentiation, and NOD-like receptor signaling. Specific genes, such as ITGB6, ITGB8, and YWHAH (PI3K-Akt signaling) and SPP1, JUN, and FOS (Toll-like receptor signaling), are significantly upregulated in the loop of Henle. Principal cells and intercalated cells in IgAN demonstrate increased expression of genes such as NFKBIA, TXNIP, CXCL3, and CXCL2 [26]. These findings underscore the complex molecular alterations occurring in different renal tubular cells of patients with IgAN, emphasizing their role in disease pathology and progression.

#### 4.2.5. Immune Cell Infiltration in the Kidney

An increased number of immune cell clusters, such as macrophages, monocytes, and dendritic cells, were identified in the kidney samples from patients with IgAN compared to those from healthy controls, while the detection of T cells was limited, which was likely due to technical challenges or low cell numbers [28]. The proportion of CD4^+^ T-cell and B-cell infiltration was negatively correlated with eGFR and positively correlated with albuminuria and glomerulosclerosis severity according to multiplex immunohistochemistry staining [30]. In renal macrophages, there was a lower expression of GPX3 and FAM49B, which protects mitochondrial function as well as combat oxidative stress. FCGBP, a negative regulator of epithelial-to-mesenchymal transition (EMT) and inflammation, also decreased, indicating potential dysregulation in immune responses [26]. Additionally, CCL2 and CX3CR1 were highly expressed. Notch signaling, glycolysis, and fatty acid and amino acid metabolism were highly enriched in IgAN macrophages, indicating the crucial role of metabolic reprogramming in macrophage activation [25]. Dendritic cells expressed high levels of genes related to HLA in kidney tissue and circulation [31]. 

### 4.3. Intercellular Crosstalk Revealed by scRNA-Seq

Interaction maps revealed a reduction in outgoing and incoming interactions of peripheral blood NK cells, whereas monocytes showed increased levels in both in IgAN. Classical monocytes became the major receiver of MIF, LIGHT, ICAM, and TGFβ and the sender of IFN-II and SELPLG, indicating that communication signaling between NK cells and monocytes was greatly modified [27]. Tfh cells were found to regulate B-cell differentiation via CD40-CD40L; in renal tissue of nephritis, results indicate that Tfh cells (CD4^+^ICOS^+^) and B cells directly interact in situ in the glomerulus [30], with intensified signaling through LTA/TNFSF14 (LIGHT)-TNFRSF14 [27]. 

As for the traditional innate cells in the kidney, MCs exhibit increased interactions with ECs (Jagged/Notch, ANGPT1/TEK, Spp1/S1pr1), proximal tubule cells (Sele/Itga5), macrophages (CSF1/IL-34/CSF1R and integrin subunit alpha X/integrin subunit alpha M/C3 axes), T cells, etc., and display inflammation mediated by autocrine responses (Cxcl1, Cxcl11) in model and human IgAN [25,29]. MCs also expressed the growth factors FGF2 and PDGFD, interacting with their respective receptors FGFR1 and PDGFRB on podocytes and the loop of Henle [26]. Cross-species analysis identified interactions between MCs and blood monocytes/macrophages via CXCL12/CXCR4: the specific blockade of the pathway significantly alleviated inflammatory injury, fibrosis, and renal function decline in mouse BAFF models. Resident macrophages further enhanced MC activation via PDGFB or C3 in situ, participating in IgAN progression [32]. Furthermore, interactions involving intercalated cells with other renal cells were notably decreased compared to normal tissue [25]. A graphical model of crosstalk between peripheral blood and renal cells in IgAN is shown in Figure 1.

## 5. Summary and Prospects

ScRNA-seq technology holds immense potential in the study of kidney diseases, which allows researchers to delve deeply into the gene expression profiles of different cell types and compare them between healthy and diseased states. Here, the novel contributions of IgAN scRNA-seq are summarized: (1) ScRNA-seq reveals the dynamic landscape of immune cells, including the activation of B cells (pathway enriched in viral infection) and Tfh cells, the active afferent and efferent signals of monocytes with (especially interferon induction gene), and the weakened cytotoxicity of CD8^+^ T cells (both in circulation and renal tissue) in IgAN patients. (2) ECs and proximal tubular cells were strongly affected in the early stage, and the role of ECs in immune cell recruitment, as well as several intraglomerular paracrine pathways involved in promoting inflammation, was identified. (3) MCs are possibly the core initiator of kidney injury. They highly express JCHAIN and facilitate the deposition of IgA immune complexes in the mesangial region. Altered (direct or indirect) cell-to-cell communication between MCs and ECs, monocytes/macrophages (recruitment, infiltration, metabolic reprogramming associated with proliferation and inflammation), T cells, podocytes, and tubular cells contribute to the onset and progression of IgAN. (4) Fibrosis with EMT features has been observed in podocytes and tubular cells. (5) The high expression of CCL2 in most renal cells (MCs, podocytes, ECs, macrophages, etc.) strongly suggests that it is the core molecule of renal progression. Studies have shown that urinary exosomal CCL2 mRNA could be a biomarker reflecting active histologic injury and renal function deterioration in IgAN [35]. The current promising targets in each cell type in IgAN identified using scRNA-seq are shown in Table 2.

Several interesting aspects require further investigation. First, more focus should be placed on the landscape of B cells, as they are centrally responsible for producing pathogenic Gd-IgA1. Recent studies have identified a subset of peripheral blood surface-/membrane-bound (mb)-Gd-IgA1^+^ B cells. Compared to healthy controls, the peripheral blood of IgAN patients was enriched with λ^+^ mb-Gd-IgA1^+^, CCR10^+^, and CCR9^+^ cells [36]. Increased levels of circulating gut-homing Bregs, memory B cells, and IgA^+^ memory B cells support the pathogenic role of intestinal mucosal hyperresponsiveness in IgAN patients [37]. Additionally, abnormal IgA class-switched CD27^−^CD21^+^ B cells, which co-express Gd-IgA1 with IgA plasmablasts, were identified in IgAN [38]. The current scRNA-seq data provide many clues about B cells in IgAN patients, such as a specific subpopulation of B cells that shows an overall decrease in the level of NF-κB activation strongly associated with IgAN progression. However, these data fail to identify the B-cell subcluster responsible for the elevation of Gd-IgA1 or fully explain the mechanism of Gd-IgA1 production. Additional methods (cytometry by time of flight, for example) that can facilitate real-time analysis at the single-cell level need to be established.

Second, monocyte-mediated chemotaxis is impaired in IgAN circulation, while macrophages show increased infiltration in the kidney with active chemokine secretion. This seeming paradox may be explained by the overwhelming kidney tissue inflammation producing chemokines locally, which can still attract and activate resident macrophages or recruit them from nearby blood vessels even if systemic monocyte recruitment is suboptimal. Pawluczyk I et al. [39] revealed the presence of macrophages juxtaposed to collecting ducts and within their lumina. In vitro, collecting duct epithelial cells exposed to macrophage-conditioned media generated from IgA1-stimulated human monocyte cell lines exhibited markedly increased expression of neutrophil-associated gelatinase and the proinflammatory cytokines IL-1β, tumor necrosis factor-α, IL-6, and IL-8 compared with non-IgA-stimulated conditions, driving tubulointerstitial inflammation and fibrosis. In addition, higher levels of glomerular CD206^+^ macrophage infiltration were reported to be associated with an increased probability of responding to immunosuppressive therapy in patients with IgAN who were at high risk of progression [40]. The above results indicate that the degree of renal macrophage infiltration may help doctors make clinical decisions, while the process of monocyte chemotaxis from the circulation to the kidney needs further study.

Third, the functional diversity of mesangial cells is gradually being discovered. These cells can not only trigger local inflammatory responses upon the binding of IgA immune complexes to receptors on their surface (such as Fc receptors) but also transform into the mesenchymal phenotype and promote fibrosis and even secrete or phagocytose IgA. Mesangial cells have been proven to be critical renal intrinsic cells with phagocytic function, capable of degrading Gd-IgA1 immune complexes after endocytosis [41,42]. A recent study reported that chemokine signal transduction and Fcγ receptor-mediated phagocytosis are canonical pathways overrepresented in IgAN glomeruli [43]. Spatial transcriptomic techniques at the single-cell level also revealed that mesangial cells were significantly enriched for cell surface/adhesion molecules and gene expression associated with vascular development or the extracellular matrix. This technique may contribute to dissecting structure-specific pathophysiology and molecular changes in IgAN [44]. 

To date, rare cell subtypes, cell differentiation trajectories, cell–cell interactions, core pathogenic genes, and major cell types associated with IgAN have been successively identified, which, to some extent, help guide the treatment of IgAN. While different scRNA-seq platforms can lead to variability in the results attributed to factors such as sequencing depth, the efficiency of capturing transcriptomic diversity, and the method of cell isolation [45]. Previous studies have compared scRNA-seq platforms and found that although technical variations exist, each platforms provide complementary strengths for different experimental subjects or goals [46]. As such, researchers should carefully select a platform based on their experimental needs and report the reason for the choice in order to ensure consistency and reproducibility across studies. It is also important to note that while each platform has its strengths and weaknesses, integration of data from different platforms is increasingly possible with advancements in computational methods for data harmonization.

The 2024 KDIGO guidelines highlight the importance of addressing atopic drivers of nephron loss in IgAN patients, including reducing the levels of pathogenic IgA and IgA immune complexes through B-cell/plasma cell depletion or the regulation of cell function and reducing glomerular inflammation. The modified form of the enteric-coated budesonide capsule (Nefecon) [47], which is now available in China, works by targeting intestinal mucosal B cells. With ScRNA-seq, the specific role of budesonide in the regulation of the immune response, particularly in specific immune subpopulations (such as Th17 cells and B-cell subpopulations), could effectively be identified. With various emerging drugs for IgAN being widely used in clinical practice [48], scRNA-seq technology can help us to understand the specific mechanism of IgAN treatment at the single-cell level to optimize its therapeutic effect. By exploring genetic signatures in different patients, researchers are able to achieve more personalized treatment strategies that maximize drug efficacy and reduce side effects. 

Challenges in scRNA-seq remain, as fresh tissue selection and the meticulous preparation of high-quality single-cell suspensions are imperative. In addition, tissue digestion during the procedure may lead to the selective loss of certain cell populations or induce the expression of stress genes, thereby engendering spurious alterations in cellular transcriptional profiles. Additionally, due to the large volume of data generated, sophisticated algorithms for data normalization, dimensionality reduction, and cell clustering are required for scRNA-seq analysis. Multi-center, large-sample sequencing data are required, and comparability and confounding factors ought to be strictly balanced among groups. The integration of single-cell technology with traditional or other omics methods (transcriptomics, proteomics, epigenomics, etc.) is not only the current focus but also the development trend [49,50]. For example, the authors of one study inferred the dysregulation of mitochondrial, lysosomal, and protein reabsorption processes in chronic kidney disease of unknown etiology via the urinary proteome [51]. The combination of urinary proteomics and renal scRNA-seq analysis can help in developing biomarkers that reflect the complex molecular biological activity of the kidney and disease activity [52,53]. ScRNA-seq continues to evolve with ongoing innovations, including emerging technologies such as the single-nucleus assay for transposase-accessible chromatin with high-throughput sequencing (snATAC-seq) [54], single-nucleus RNA sequencing (snRNA-seq) [55], and spatial transcriptome sequencing [56]. With the advancement of technology and the reduction in sequencing costs, scRNA-seq is poised to play a greater role in early diagnosis, disease surveillance, and personalized therapy for renal diseases, driving advances in precision medicine. 

## 6. Conclusions

In summary, the application of scRNA-seq to renal tissue and blood offers unprecedented opportunities to dissect cellular heterogeneity, unravel disease mechanisms, and identify therapeutic targets in IgAN with high precision and resolution. This technology holds immense promises for advancing our understanding and management of kidney diseases.

## Figures and Tables

**Figure 1 biomolecules-15-00191-f001:**
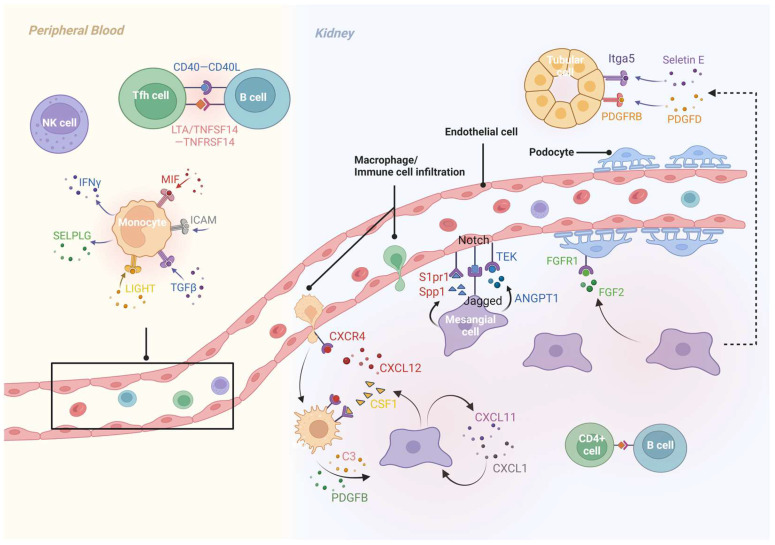
Peripheral blood and renal cell crosstalk in IgAN from review of scRNA-seq. Intercellular communication analysis of IgAN showed enhanced interaction between Tfh cells and B cells both in peripheral blood and kidney, peripheral blood monocytes are active in receiving and transmitting signals. Crosstalk established between mesangial cells, podocytes, endothelial cells, tissue immune cells and tubular cells further amplifying inflammatory responses both locally and remotely. (Graph created with Biorender.com).

**Table 1 biomolecules-15-00191-t001:** Summary of studies in IgAN applying scRNA-seq technology.

Authors/Publication/Year	PMID	Deposited Data	Platform	Organisms	Included Sample (Normal vs. IgAN)	Findings
Zheng Y, et al. Cell Rep. 2020 [25]	33357427	Gene Expression Omnibus: GSE127136/BIG Submission Portal: HRA000342	Illumina Hiseq-4000	Human	6 vs. 13 (Kidney); 5 vs. 5 (CD14^+^ PBMCs)	Upregulation of JCHAIN, WFDC2 and genes related to ECM accumulation in mesangial cells; Upregulation of CX3CR1 and CCL2 in KRMs and type I interferon-encoding genes in CD14^+^ PBMCs in IgAN; Specific gene expression patterns in KRMs and CD8+ T cells suggest abnormal regulation associated with proliferation and inflammation; A transitional type of ICs/PCs was identified with fibrotic characteristics, involving Wnt, FGF signaling;
Tang R, et al. Front Immunol. 2021 [26]	33936064	Gene Expression Omnibus: GSE171314	Illumina HiSeq X10	Human	3 vs. 4 (Kidney)	Identification of novel genes upregulated in IgAN mesangial cells (such as MALAT1, GADD45B, SOX4, and EDIL3) associated with cell proliferation and matrix accumulation; Enrichment of inflammatory pathways (including TNF signaling, IL-17 signaling, and NOD-like receptor signaling) in tubule cells of IgAN patients; Observed an upregulation of genes related to fibrosis and immune cell recruitment (such as FGF2, PDGFD, CXCL1, and CCL2) and an elevated number of macrophages, monocytes, and dendritic cells in kidney samples of IgAN subjects.
Zeng H, et al. Cell Biosci. 2021 [27]	34895340	BIG Submission Portal: HRA000831	BD Rhapsody	Human	6 vs. 10 (PBMC)	The number and cytotoxicity of NK cells in IgAN patients was reduced and negatively correlated with clinical parameters like UPCR, IgA, and Gd-IgA1; DEGs in B cells were enriched in viral infection pathways, B cell subset with suppressed NF-κB signaling positively associated with disease progression; Monocyte subset expressing interferon-induced genes was positively associated with the clinical severity of IgAN.
Chen Z, et al. J Cell Mol Med. 2021 [28]	33754492	NA	Illumina HiSeq	Human	1 vs. 3 (Kidney)	Upregulation of CLIC1 and RPS26 in mesangial cells and JUNB in podocyte of IgAN.
Zambrano S, et al. Kidney Int. 2022 [29]	34968552	Gene Expression Omnibus: GSE166793	Illumina HiSeq-3000	gddY Mice	5 vs. 5 (Glomeruli)	Endothelial cells participated in the early pathogenesis of IgAN, particularly in recruiting and infiltrating immune cells into the glomerulus; Paracrine pathways involving mesangial cell-derived Slit3 potentially activating receptors in podocytes/endothelial cells; Proximal tubular cells were strongly affected at the early stage and potential glomerular-tubular crosstalk.
Du W, et al. Front Immunol. 2022 [30]	35983053	CNGB Sequence Archive: CNP0002798	Illumina NovaSeq	Human	4 vs. 3 (PBMC)	A significant higher proportion of Th2, Th17 and Tfh cells were observed in IgAN, the proportion of Tfh cells was positively correlated with the severity of proteinuria; Infiltration of CD4^+^T and B cells in kidney were increased and positively correlated with IgAN renal damage; Enhanced interactions between Tfh cells and B cells mediated by the TNFSF14-TNFRSF14 pathway;
Chen Q, et al. Front Immunol. 2023 [31]	37908345	NA	Illumina NovaSeq 6000; Illumina HiSeq X10	Human	3 vs. 3 (PBMC) 3 vs. 4 (Kidney)	Increased level of CCR2 in the classical, intermediate, and non-classical monocytes in NS-IgAN; Treg2 (CCR7low, TCF7low, and HLA-DRhigh) expressed significant levels of CCR4 and GATA3; Increased levels of CCL2, PRSS23, and genes related to epithelial-mesenchymal transition in podocytes of NS-IgAN; PTGDS is significantly downregulated following podocytes injury;
Chen X, et al. JCI Insight. 2024 [32]	38716725	Reproduce analysis from GSE127136 and GSE166793	Illumina HiSeq	Human, gddY mouse	Human: 3620 cells (Blood, Kidney); Human and gddY IgAN cross-species analysis: 3076 cells (Glomeruli)	Mesangial cells highly expressed inflammatory markers (such as CXCL12, CCL2, CSF1, and IL-34) specifically interacted with macrophages through various pathways, including CXCL12/CXCR4, CSF1/IL-34/CSF1 receptor, and integrin subunit alpha X/integrin subunit alpha M/complement C3 axes; IgAN macrophages largely derived from infiltrating blood monocytes and expressed high levels of CXCR4 and PDGFB; Levels of CXCL12, C3, mannose receptor C-type 1, and CD163 proteins were negatively correlated with eGFR value and were prognosis indicators in IgAN; Specific blockade of the CXCL12/CXCR4 pathway substantially attenuated inflammatory injury, fibrosis, and renal function decline in a mouse model of IgAN.

**Table 2 biomolecules-15-00191-t002:** Summarizing the promising candidate genes in each cell type.

Cell Type	Cell Subtype	Key Related Gene
CD8^+^ T cells	—	FCGR3A, GZMB, KLRD, FGFBP2, GZMH [25]; PFR1, NKG7, GZMA, GZMB, GZMH, GNLY [27]
NK cells	3 subsets [27]	HLA-C, FOSB, JUNB, PFR1, NKG7, GZMA, GZMB, GZMH, GNLY [27]
CD4^+^T cell	7 subsets: Naïve CD4^+^T cell (CCR7^+^SELL^+^), Th1 (TBX21^+^CXCR3^+^), Th2 (GATA3^+^CCR4^+^), Th17 (RORC^+^CCR6^+^), Tfh (CXCR5^+^ICOS^+^), Treg (IL2RA^+^FOXP3^+^)and others [30]	ICOS, STAT3, IL21R, IL6R, LTA, TNFSF14 [30]
Monocytes	4 subsets [27]; 3 subsets: Classical monocytes (CD14^++^CD16^−^), Intermediate monocytes (CD14^++^CD16^+^), and Non-classical monocytes (CD14^+^CD16^++^) [31]	IFI44, IFI44L, IFI6, and ISG15 [25]; S100A8, S100A9, CXCL8, HLA-C, TMEM176A, TMEM176B, HLA-B, NAMPT, NDUFS3, TNFRSF1A, CCL3, CCL4, CCL3L3, CCL4L2 [27]; VSIG4, HLA-DPA1, HLA-DPB1, CCR2 [31]
B cells	4 subsets [27]	HLA-C, CD83, CD69, NFΚBIA, YPEL5, DNAJA1, HLA-DPB1, SWAP-70, FCRLA, SPIB, TNFRSF13C [27]
Mesangial Cells	3 subtypes: proliferative MCs (marked by Mki67, Pdgfrb), MCs (marked by Pdgfrb), and inflammatory MCs (marked by Pdgfrb, Cxcl12, Csf1, Il34, Cxcl16) [32]	JCHAIN, THY1, WFDC2, SPP1, KNG1, PLGRKT, CCL2, AOC3 [25]; CXCL12, IL34, CSF1, CCL2, -3, and -4, PDGFRB, ITGAX/ITGB2 [32]; Cd36, Stab2, osteopontin, serpins, Slit3 [29]; MALAT1, GADD45B, SOX4, EDIL3, FOS, ID2, MT-RNR1 [26]; CCL2, CFH, PRSS23 [31]
Endothelial Cells	3 subtypes: GECs (Ehd3^+^), Afferent arteriolar endothelial cells (Pecam1^+^ and Ehd3^−^/Gja5^+^/Adora^+^), Efferent arteriolar endothelial cells (Pecam1^+^ and Ehd3^−^/Gja5^−^/Adora^−^) [29]	KDR, FLT1 [25]; Edn1, f8, Sele, Vcam1, Icam1, Spp1, Cd36, Nostrin, Cx3cl1, Cxcl1, Cxcl11, Serpina, MHC class I, Ltbp1, Cxcl10) [29]; SOX4, MT-RNR1, PECAM1, UTRN, MTATP6P1, MT-ND4L, SELP, XIPS [26]; CCL2, TGFB1, COL1A1 [31]
Podocytes	—	FXYD5, CD74, B2M [28]; C3, B2m, H2-k1, Col1a2, Acta2 [29]; PRSS23, NGF, HES1 [26]; TGFB1, CAV1, TAGLN, CCL2 [31];
Kidney macrophage	2 subsets [29]	CCl2, CX3CR1 [25]; MHC II, Cd36, PPARγ, caveolin-1 [29]; CD68, IL1B, CD86, PDGFB, TNF, C3, C1QC, IFI6/44/44L, LUM, COL1A2, CXCL2, -6, and -11, CX3CR1 [32]
Proximal tubule cells	5 subtypes [28]	SLC5A12, CUBN [25]; C3 [29]; MMP7, MYC [28]; TNF signaling, IL-17 signaling, NOD-like receptor signaling [26,31]
Loop of Henle cells	—	SLC12A1 [25]; ITGB6, ITGB8, YWHAH, SPP1, JUN, FOS [26]
Principal cells/Intercalated cells	—	NFKBIA, TXNIP, CXCL3, CXCL2 [26]

## Data Availability

The datasets presented in this study can be found in online repositories.

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
