# Peer review of "Single-Cell RNA-Sequencing Analysis Provides Insights into IgA Nephropathy"

_biomolecules, 2025, doi:10.3390/biom15020191_

Round 1
Reviewer 1 Report
Comments and Suggestions for Authors
Single cell technology is now widely used in research and molecular mechanisms involved in IgA nephropathy have not yet been explored in depth. This manuscript reviews all information the authors could collect about single cell transcriptomics from kidney and blood in IgA nephropathy. The idea of a review on this topic is great, but I have major concerns regarding the validation of a big part of what is described due to data accessibility barriers and missing associated literature.
Major comments:
- Table 1: must be particularly revised. Specially: the authors cannot select the authors they list, they should report in the order the authors are listed in publications. Including the author names and year is not enough: the exact references need to be added. Please separate each reference with a line so that the “findings” make sense. What means “KRMs” ? the word “Resources” does not make sense for specifying the organism studied (human or mouse). The second reference is even wrong: the GSE131685 is for 3 controls from other authors. The Zambrano reference includes a wrong year, Smart-seq2 is a method not a platform, samples were not whole kidneys but mouse glomeruli. Despite what is claimed, several datasets are barely accessible to the scientific community and/or do not have associated publication, making the findings not verifiable. The authors also do not really try to cross reference some findings across the studies. The GSE127136 is not mentioned, even though its quality/rigor is highly doubtful. The last reference is not accessible and does even not mention what samples were used.
In summary, this entire manuscript is based on Table 1 and contain data that cannot be verified/are not trustable. Unreliable information/ conclusions cannot be dispersed to the scientific community. So far there are only few “real” single cell datasets available for IgA nephropathy in the world.
Detailed comments:
- Abstract: The sentence “The initial role of mesangial cells …” needs correction. Last sentence: please update with “complex molecular mechanisms of IgAN”.
- Introduction: second sentence: although IgAN is one of the most important kidney diseases in China, I would also mention other countries including USA. Last sentence of the first paragraph, I would include the microbiome. Sentence “while there are some limitations …”: please include how. The sentence “Given the diverse roles …” is not clear. Sentence: “Its strengths …”, please include cell-cell communication.
- Section 2 (General mechanisms/pathophysiology): last paragraph contains a duplicated sentence.
- Title of the section 3 (“Development of scRNAseq and its application in nephrology”): this section is only about scRNAseq methods, nothing related to kidney. Al the last paragraph lacks references, the last sentence is not clear and “Tyrisin” stands for what ? trypsin ?
- Section 4: please correct the word “researches”. First sentence: “was published in 2020”, please add the reference.
- Section 4.1: first sentence, please provide the exact author name order of the reference, as in Table 1 (not selected names).
- I would separate the results from PBMCs and kidneys: I would make section 4.2. as a kidney section.
- Title of section 4.2: please replace “cell subsets” by “cell types”.
- Section 4.2.4: the first sentence does not mention that results may just be due to technical issues.
Comments on the Quality of English Language
The English needs to be improved (vocabulary, structure of sentences).
Author Response
Single cell technology is now widely used in research and molecular mechanisms involved in IgA nephropathy have not yet been explored in depth. This manuscript reviews all information the authors could collect about single cell transcriptomics from kidney and blood in IgA nephropathy. The idea of a review on this topic is great, but I have major concerns regarding the validation of a big part of what is described due to data accessibility barriers and missing associated literature.
Major comments:
Table 1: must be particularly revised. Specially: the authors cannot select the authors they list, they should report in the order the authors are listed in publications. Including the author names and year is not enough: the exact references need to be added. Please separate each reference with a line so that the “findings” make sense. What means “KRMs” ? the word “Resources” does not make sense for specifying the organism studied (human or mouse). The second reference is even wrong: the GSE131685 is for 3 controls from other authors. The Zambrano reference includes a wrong year, Smart-seq2 is a method not a platform, samples were not whole kidneys but mouse glomeruli. Despite what is claimed, several datasets are barely accessible to the scientific community and/or do not have associated publication, making the findings not verifiable. The authors also do not really try to cross reference some findings across the studies. The GSE127136 is not mentioned, even though its quality/rigor is highly doubtful. The last reference is not accessible and does even not mention what samples were used.
Responds: Thank you for your suggestions. We have modified Table 1 according to your suggestion, please check. KRMs is an acronym for Kidney-resident macrophages, which we have listed in our acronym table (and was reordered alphabetically). The second reference uses healthy kidney tissues of three human donors in GSE131685 and reproduces the downstream Analysis, forming GSE171314 data, we have revised in the paper. Also, we revised the Zambrano reference, platform, and samples in Table 1. As for GSE127136 you mentioned, is uploaded by Zheng Y, et al. J Cell Rep., 2020 (the first row in Table 1). They uploaded their data both to GEO and BIG Submission Portal. We have added it in Table 1 accordingly.
It is indeed difficult to reproduce or verify the huge amount of data from the ScRNA-seq study, and we are trying our best to summarize each article presented. The last reference conducted single-cell cross-species transcriptomic analysis on human IgAN, mouse (gddY) IgAN, they reproduce the analysis from data GSE127136 (Table 1 first row, their own research) and GSE166793 (Table 1 fifth row, from Sonia Zambrano), and crosstalk analysis in human IgAN and the ddY IgAN mouse model (3,076 cells). We made corrections in Table 1.
In summary, this entire manuscript is based on Table 1 and contain data that cannot be verified/are not trustable. Unreliable information/ conclusions cannot be dispersed to the scientific community. So far there are only few “real” single cell datasets available for IgA nephropathy in the world.
Responds: Indeed, there are still very few publicly available single-cell datasets for IgA nephropathy worldwide, which poses challenges. We fully understand and appreciate your concerns regarding the reliability of the data and the trustworthiness of the conclusions. Only a few of the data in Table 1 are inaccessible, but due to the huge data amounts, it is difficult for us to further qualified control, and we can only summarize the content authors trying to convey in their manuscript. We also need to be cautious in our interpretations to avoid overreaching conclusions. Thank you for your valuable suggestions, which are instrumental in improving the quality of our manuscript. We hope you can recognize our efforts and that the revised version will meet your expectation.
Detailed comments:
- Abstract: The sentence “The initial role of mesangial cells …” needs correction. Last sentence: please update with “complex molecular mechanisms of IgAN”.
Responds: We have revised it.
Introduction: second sentence: although IgAN is one of the most important kidney diseases in China, I would also mention other countries including USA. Last sentence of the first paragraph, I would include the microbiome. Sentence “while there are some limitations …”: please include how. The sentence “Given the diverse roles …” is not clear. Sentence: “Its strengths …”, please include cell-cell communication.
Responds: We have revised it in Introduction Part.
Section 2 (General mechanisms/pathophysiology): last paragraph contains a duplicated sentence.
Responds: Duplicate sentences have been deleted.
Title of the section 3 (“Development of scRNAseq and its application in nephrology”): this section is only about scRNAseq methods, nothing related to kidney. Al the last paragraph lacks references, the last sentence is not clear and “Tyrisin” stands for what ? trypsin ?
Responds: Thank you for your suggestion, Section 3 introduces the development of scRNA-seq technology and workflow, it’s discuss the application of scRNA-seq technology in nephrology from the fifth line of the second paragraph. Samples are usually from kidneys or peripheral blood, and the sample processing is briefly described. If possible, we'd like to keep the current subheading. We have corrected “Tyrisin” to trypsin.
Section 4: please correct the word “researches”. First sentence: “was published in 2020”, please add the reference.
Responds: We have corrected it and added the reference.
Section 4.1: first sentence, please provide the exact author name order of the reference, as in Table 1 (not selected names).
Responds: Due to the many authors involved in each article, we previously listed only the first author and corresponding author. Since it is unsightly to list all authors, we refer to other single-cell sequencing reviews (Blood. 2023 Jul 27; 142(4):313-324) and revised Table 1. And cited only the first author when quoting the article. Please check.
I would separate the results from PBMCs and kidneys: I would make section 4.2. as a kidney section.
Responds: Related revisions have been made.
Title of section 4.2: please replace “cell subsets” by “cell types”.
Responds: We have revised it.
Section 4.2.4: the first sentence does not mention that results may just be due to technical issues.
Responds: We revised that "On the basis of available techniques, urinary tubules are considered to consist of renal tubules (proximal renal tubules, Henle's loop, distal renal tubules) and collecting tubules."
Comments on the Quality of English Language
The English needs to be improved (vocabulary, structure of sentences).
Responds: Our manuscript has undergone English language editing by MDPI. The text has been checked for correct use of grammar and common technical terms, and edited to a level suitable for reporting research in a scholarly journal. Please check.
Reviewer 2 Report
Comments and Suggestions for Authors
This manuscript is intended as a "Research ARTICLE" but it doesn't have the structure of a scientific article. It lacks a Material and Methods or Results section, so it could be more a kind of review paper. Nevertheless, it is a plain, superficial manuscript without any deep presentation or discussion of data. scRNA.seq gives huge amounts of data and any analysis/metaanalysis should be welcome to provide new knowledge on renal diseases and highlight new possible markers, but nothing of this is achieved in this manuscript. I would suggest authors to center in one/few aspects of nephropathy onset and progression and make a much more in deep treatment of scRNA.seq data.
Author Response
This manuscript is intended as a "Research ARTICLE" but it doesn't have the structure of a scientific article. It lacks a Material and Methods or Results section, so it could be more a kind of review paper. Nevertheless, it is a plain, superficial manuscript without any deep presentation or discussion of data. scRNA.seq gives huge amounts of data and any analysis/metaanalysis should be welcome to provide new knowledge on renal diseases and highlight new possible markers, but nothing of this is achieved in this manuscript. I would suggest authors to center in one/few aspects of nephropathy onset and progression and make a much more in deep treatment of scRNA.seq data.
Responds: Thank you for your feedback. Our manuscript was uploaded as the REVIEW paper. As for your concern, we have made corresponding improvements, mainly in the "Summary and Prospect" section, including the first four paragraphs which elaborate on the new perspectives that ScRNA has brought to B cells, monocytes, mesangial cells, etc., and the fifth paragraph which focuses on the new depth that ScRNA has brought to the treatment of IgAN. Also, we have added Table 2 to present supplementary information on potential targets of different types of cells. In addition, our manuscript has undergone English language editing by MDPI. The text has been checked for correct use of grammar and common technical terms, and edited to a level suitable for reporting research in a scholarly journal. Please check. We hope you can recognize our efforts and that the revised version will meet your expectations.
Reviewer 3 Report
Comments and Suggestions for Authors
I considered the manuscript entitled “Single-cell RNA-Sequencing Analysis Provides Insights into IgA Nephropathy" by Ming Xia, et al, that is intended to be published in Biomolecules journal.
The aim of the manuscript appears interesting and new. However, as authors note there is few information in the field. Though the manuscript beguins focused and informative, and the final summary is apparently well constructed, it contains a central part that is just copy and paste of the articles that exists in the literture at this time. A copy and paste.
Should the authors expand the concept the authors of the articles try to give? in this section of the manuscript
Comments on the Quality of English Language
it can be ameliorated by the editorial correctors
Author Response
Comments and Suggestions for Authors
I considered the manuscript entitled “Single-cell RNA-Sequencing Analysis Provides Insights into IgA Nephropathy" by Ming Xia, et al, that is intended to be published in Biomolecules journal.
The aim of the manuscript appears interesting and new. However, as authors note there is few information in the field. Though the manuscript beguins focused and informative, and the final summary is apparently well constructed, it contains a central part that is just copy and paste of the articles that exists in the literture at this time. A copy and paste.
Should the authors expand the concept the authors of the articles try to give? in this section of the manuscript.
Responds:Thank you for your suggestions. As for your concern, we have made corresponding improvements, mainly in the "Summary and Prospect" section, including the first four paragraphs which elaborate on the new perspectives that ScRNA has brought to B cells, monocytes, mesangial cells, etc., and the fifth paragraph which focuses on the new depth that ScRNA has brought to the treatment of IgAN. Also, we have added Table 2 to present supplementary information on potential targets of different types of cells.
Comments on the Quality of English Language
it can be ameliorated by the editorial correctors.
Responds: Our manuscript has undergone English language editing by MDPI. The text has been checked for correct use of grammar and common technical terms, and edited to a level suitable for reporting research in a scholarly journal. Please check. We hope you can recognize our efforts and that the revised version will meet your expectations.
Round 2
Reviewer 1 Report
Comments and Suggestions for Authors
Thank you for revising this manuscript. GSE171314 is still missing in the second reference of Table 1. Please be consistent throughout the paper with citing authors: e.g. "Tong R" instead of Rong Tang, "Zeng H.", etc.
Author Response
Thank you for revising this manuscript. GSE171314 is still missing in the second reference of Table 1. Please be consistent throughout the paper with citing authors: e.g. "Tong R" instead of Rong Tang, "Zeng H.", etc.
Response:We have made corresponding revisions, please check.
Reviewer 2 Report
Comments and Suggestions for Authors
The authors have made a great effort to improve the manuscript. This is now a systematical and complete description of the possibilities of scRNA.seq to improve the knowledge of renal diseases. I found this revision timely and complete and have only two minor concerns that I would like to be addressed by the authors to improve the manuscript.
1.- There are different platforms for scRNA.seq. Do they give similar or different results or the outputs are substantially different among them? A discussion on this topic would enrich the manuscript.
2.- I would like to see the bioinformatics section developed more in deep (now it is just one reference). The authors should include commertial platforms for data analysis or programs used to acquire and prune data, to compare with the reference genomes etc.
Author Response
1.- There are different platforms for scRNA.seq. Do they give similar or different results or the outputs are substantially different among them? A discussion on this topic would enrich the manuscript.
Response: We appreciate the reviewer’s comment regarding the use of different scRNA-seq platforms. Indeed, there is a growing variety of platforms available for scRNA-seq including Fluidigm C1 platform, 10x Genomics chromium platform, BD rhapsody, et al. While different scRNA-seq platforms can lead to variability in the results, the results obtained are not contradictory, and the discrepancy may attributed to factors such as sequencing depth, the efficiency of capturing transcriptomic diversity, and the method of cell isolation. For example, platforms like 10x Genomics offer high throughput and greater scalability, while Fluidigm C1 platform (using SMART-seq/C1 method) is often preferred for its deeper transcript coverage at the cost of lower throughput. Previous studies have compared scRNA-seq platforms and found that although technical variations exist, each platforms provide complementary strengths for different experimental subjects or goals (Genome Med. 2017 Aug 18;9(1):75; Front Physiol. 2021 Oct 13:12:752679.), and researchers need to carefully choose the platform that best suits their experimental goals. We revised section 5 (Summary and prospects) paragraph 5, please check.
2.- I would like to see the bioinformatics section developed more in deep (now it is just one reference). The authors should include commertial platforms for data analysis or programs used to acquire and prune data, to compare with the reference genomes etc.
Response:Thank you for your constructive feedback. In response to your comment, we added related discussion on both commercial platforms and commonly used programs in section 3 (Development of scRNA‑seq technique and its application in nephrology), first paragraph.
Reviewer 3 Report
Comments and Suggestions for Authors
It has clearly improved after the profound correction
Author Response
Thank you for your support.